# The application of CBL and mind mapping combined with Mini-CEX teaching mode in the cultivation of clinical competence of ultrasound residents

Rong Liu◉*, Lei Li, Ying Zhou, Lu Xiong

Department of Medical Ultrasound, Tianyou Hospital Affiliated to Wuhan University of Science and Technology, Wuhan, Hubei, China

* 865186639@qq.com

## Abstract

### Objective

This study aims to evaluate the efficacy of the Mini Clinical Evaluation Exercise (Mini-CEX) in conjunction with Case-Based Learning (CBL) and mind mapping methodologies in fostering the clinical competencies of ultrasound residents.

### Methods

A cohort of forty-two general practitioners who underwent standardized training at the Department of Ultrasound Medicine at Tianyou Hospital, affiliated with Wuhan University of Science and Technology, from December 1, 2022 to December 1, 2024, were selected for this research. These practitioners were randomly assigned to either an experimental group or a control group using a random number table. The experimental group engaged in training utilizing the Mini-CEX combined with CBL and mind mapping approach, whereas the control group adhered to a traditional training regimen. Subsequently, comparative analyses were conducted on theoretical and practical test scores as well as clinical teaching satisfaction ratings between the two groups. Additionally, variations in Mini-CEX evaluation scores for the experimental group were assessed at admission, after two weeks, and after four weeks of training.

### Results

The training approach that integrated Mini-CEX with CBL and mind mapping demonstrated significantly superior outcomes compared to the traditional training method regarding theoretical and practical performance, as well as teaching satisfaction, with statistically significant differences noted ($p < 0.05$). A comparison of Mini-CEX scores among 21 general practitioners at two and four weeks post-training versus admission yielded statistically significant results ($p < 0.05$).

**Data availability statement:** All relevant data are within the manuscript and its Supporting Information files.

**Funding:** The author(s) received no specific funding for this work.

**Competing interests:** The authors have declared that no competing interests exist.

## Conclusion

The amalgamation of Mini-CEX with CBL and mind mapping methodologies markedly enhances the instructional effectiveness of standardized resident training.

## Introduction

The standardized training of residents is a pivotal component of the medical education framework in China, aimed at cultivating proficient clinicians equipped with sound clinical reasoning and practical skills [1].Given the extensive utilization of ultrasound technology, the training of resident ultrasound practitioners is of paramount importance. Nonetheless, the development of clinical competencies during ultrasound residency encounters several challenges. Residents frequently express concerns regarding the brevity of ultrasound learning periods, the vastness of required knowledge, the complexity of operations, and the propensity for misdiagnoses or overlooked diagnoses. Consequently, the effectiveness and relevance of educational strategies are critical in residency training. As medical education continues to evolve, conventional teaching methods are increasingly inadequate in fulfilling contemporary educational demands for skill development among students [2]. In response to these challenges, our department has implemented a case-based teaching approach (CBL) in conjunction with mind mapping techniques during resident training. Through the examination and discussion of real clinical cases, we encourage active learning and the development of systematic thinking skills among students. The construction of mind maps enables the organization and consolidation of disease-related knowledge [3], thereby facilitating a more comprehensive understanding and retention of intricate medical information. Furthermore, addressing the observed deficits in humanistic communication skills, teamwork capabilities, and operational competencies among residents, we have incorporated the Mini Clinical Evaluation Exercise (Mini-CEX) into our training framework, And it is improved according to the characteristics of the ultrasound specialty. The Mini-CEX facilitates direct observation for evaluating the clinical skills of residents by monitoring their interactions with patients and colleagues, procedural execution, and adherence to standard protocols during ultrasound examinations, thereby providing immediate feedback and guidance [4].This study adopted a randomized controlled trial, aiming to investigate the impact of integrating Mini-CEX with Case-Based Learning (CBL) and Mind Map training methodologies on the education of ultrasound residents. Additionally, it sought to assess the effectiveness of this combined approach in enhancing the clinical competencies of the residents, thereby offering innovative strategies for ultrasound residency training.

## Subjects and methods

### Participants

A total of forty-two general practitioners, consisting of 23 males and 19 females, were selected for this study. These participants underwent standardized residency

training in the Department of Ultrasound Medicine at Tianyou Hospital, which is affiliated with Wuhan University of Science and Technology, from December 1, 2022 to December 1, 2024. The cohort comprised 14 students from the 2021 intake, 15 from the 2022 cohort, and 13 from the 2023 group, all of whom successfully passed the hospital's recruitment examination. This group included resident doctors, social training participants, and graduate students with four certifications. The forty-two residents were randomly categorized into two groups: a control group (n = 21), comprising 11 males and 10 females with a mean age of (24.10 ± 1.78)years, which included 7 students from 2021, 6 from 2022, and 8 from 2023, who received traditional teaching and training methods (average age 23.90 ± 2.17 years), also distributed evenly across the three grades. No significant differences in age or sex were observed between the groups(p > 0.05).

Inclusion criteria:(1) Postgraduate students are full-time first-year students majoring in clinical medicine; (2) Undergraduate students and socialized standardized training trainees are full-time medical graduates with a bachelor's degree in clinical medicine or related fields. (3) Resident physicians are those who have not yet participated in the standardized training for resident physicians and have not yet been promoted to intermediate technical titles; (4) Informed consent for the research. Exclusion criterion: Failure to cooperate with the researcher.

This study was approved by the Ethics Committee of Tianyou Hospital Affiliated to Wuhan University of Science and Technology (Approval No. LL-2022-11-02-02)., Written informed consent was obtained from all participants, ensuring their understanding of the study's purpose and procedures.

## Study methods

The training period for both groups of trainees was 4 weeks. After entering the department, each trainee was assigned a mentor. During the training period, they all participated in small lectures in the department, discussions on difficult cases, professional learning, and ultrasound operation training. All teaching was strictly implemented in accordance with the detailed rules and contents of the resident training of the Ultrasound Medicine Department, combined with the characteristics of this major. The control group received conventional residential teaching. That is, after a resident physician enters the department, a supervising teacher is designated. According to the requirements of the standardized training teaching syllabus for resident physicians, the method of explaining while working in clinical work is replaced with integrated instruction. After resident physicians have a certain foundation in ultrasound medical operations, they operate on the machine under the guidance of the teaching instructors. an exit examination is conducted after the learning is completed. The experimental group employed a combination of CBL and Mind Map methodologies to merge ultrasound diagnostics with clinical and basic medical education into a cohesive teaching framework. Prior to lectures, the instructional team provided the study group with case studies, pertinent data, and ultrasound images, facilitating a review of relevant anatomical and diagnostic knowledge. Students were encouraged to conduct analyses, formulate diagnoses, and differentiate cases, culminating in the creation of mind maps. In the classroom, instructors elucidated the various ultrasound images and related clinical data changes throughout the case progression, elucidating concepts such as "one map for multiple diseases" and "multiple maps for one disease" in the context of ultrasound medicine, Reduce students' confusion and doubts about the learning of such ultrasound images, Instructors guided students in utilizing thought maps to systematically organize and reinforce learned material, thereby fostering clinical reasoning skills and enhancing ultrasound diagnostic capabilities, achieving a pedagogical outcome that effectively integrates foundational diagnosis with clinical practice. Throughout the training, resident physicians participated in hands-on operational practice, during which instructors detailed procedural steps, techniques, and safety considerations, enabling students to proficiently select probes, conduct pre-examination preparations, and adjust real-time ultrasound images while adhering to standard anatomical sections. Emphasis was also placed on the importance of communication with patients and colleagues, alongside conducting appropriate physical examinations to nurture the practical skills of resident doctors.

 

## Evaluation metrics

The efficacy of the training was assessed using the Mini-CEX tool at three intervals: upon admission, two weeks post-training, and four weeks post-training. The improved Mini-CEX evaluations were tailored to the professional competencies specific to ultrasound, assessing the residents' abilities in routine ultrasound procedures. Feedback was concentrated on six key areas: medical history taking, ultrasound examination, doctor-patient communication, clinical reasoning, organizational efficiency, and overall performance. In 1999, the Accreditation Council of Graduate Medical Education (ACGME) stated that qualified residents should have patient care and medical knowledge, and practical skills. Practical learning and improvement; interpersonal relationship and communication skills (KIUS) professionalism, and the six core competencies of system-based practice. The Mini-CEX assessment mainly evaluates the abilities of trainees in aspects such as interpersonal communication, humanistic care, professional ability, professional spirit. Therefore, the assessment content of the improved Mini-CEX is aligned with these six core competencies. The Mini-CEX scoring system employed a scale where scores ranging from 1 to 3 were deemed insufficient, 4–6 were categorized as qualified, while those scoring between 7 and 9 were deemed excellent. A structured questionnaire was developed to assess the conventional teaching approach in conjunction with the Mini-CEX, CBL, and Mind Map instructional techniques. The focal points of the questionnaire encompassed: stimulating learner interest, enhancing critical thinking and problem-solving abilities, fostering clinical reasoning, improving communication competencies, bolstering confidence in clinical learning, and augmenting proficiency in practical skills, all of which align with standardized training requirements. The survey was administered to 42 residents, each of whom was tasked with completing a questionnaire. A total of 42 questionnaires were distributed, and all were returned, resulting in a complete response rate of 100%.

## Statistical methods

The statistical analysis was conducted using SPSS 22.0 software. It was confirmed that the data adhered to a normal distribution; thus, the measurement data were presented as mean ± standard deviation (±s). An independent sample t-test was employed for inter-group comparisons, with a significance threshold set at $p < 0.05$.

## Results

The results indicated that the examination scores for both theoretical and practical components in the Mini-CEX combined with CBL and Mind Map instructional approach surpassed those recorded in the traditional teaching method, with the differences reaching statistical significance ($p < 0.05$), refer to Table 1 for details.

Mini-CEX scores revealed that the experimental group's performance at 2 weeks and 4 weeks post-training significantly exceeded their initial scores at admission, with differences noted as statistically significant ($p < 0.05$).see Table 2 for details. The reliability analysis of the questionnaire yielded a Cronbach's Alpha of 0.812, indicating substantial internal consistency. The validity assessment, measured by the Kaiser-Meyer-Olkin (KMO) statistic of 0.754, alongside Bartlett's test of sphericity yielding a P value of 0.000, confirmed robust content validity. refer to Table 3 for detailed scores. The questionnaire

**Table 1. Comparative analysis of mean scores between the experimental group and control group (scores, x±s).**

| group | Theory test score | | Practical test score | |
|---|---|---|---|---|
| | Entrance score | Graduation score | Entrance score | Graduation score |
| control group | 70.00±4.15 | 84.05±3.35 | 67.71±4.22 | 83.52±3.30 |
| experimental group | 69.52±4.64 | 90.05±3.20 | 68.14±3.15 | 89.81±3.27 |
| t value | 0.35 | 5.93 | 0.37 | 6.21 |
| P value | >0.05 | <0.05 | >0.05 | <0.05 |

**Table 2. Mini-CEX score comparisons for the experimental group across different time intervals (points, x±s).**

| item | Upon admission | 2 weeks | 4 weeks | 95%CI | P value |
|---|---|---|---|---|---|
| History inquiry | 5.19±0.63 | 6.66±0.55 | 8.05±0.29 | 4.90-5.48 | <0.001 |
| Ultrasonic examination | 3.18±0.41 | 5.25±0.69 | 7.58±0.34 | 2.99-3.36 | <0.001 |
| Doctor-patient communication | 4.81±0.43 | 6.14±0.30 | 7.78±0.37 | 4.63-5.03 | <0.001 |
| Clinical thinking | 3.56±0.6l | 6.00±0.42 | 7.62±0.44 | 3.25-3.80 | <0.001 |
| Organizational effectiveness | 4.24±0.80 | 5.86±0.40 | 7.79±0.50 | 3.88-4.60 | <0.001 |
| Overall evaluation | 4.08±0.75 | 5.81±0.54 | 7.82±0.52 | 3.74-4.42 | <0.001 |

**Table 3. The results of the validity analysis of the questionnaire survey.**

| Kaiser-Meyer-Olkin Measur the appropriateness of sampling | | 0.754 |
|---|---|---|
| Bartlett's sphericity test | Approximate chi-square | 98.971 |
| | df | 21 |
| | sig | 0.000 |

findings revealed that the experimental group exhibited superior satisfaction levels with the instructional methods compared to the control group, with the difference achieving statistical significance ($p < 0.05$), refer to Table 4 for detailed scores.

## Discussion

In the educational framework of clinical specialties, the allocation of class hours dedicated to ultrasonic diagnosis is notably limited, resulting in a suboptimal comprehension of ultrasonic medical principles among students. Conventional pedagogical approaches fall short in igniting students' critical thinking and effectively consolidating their acquired knowledge, leading to inadequacies in their ability to tackle clinical challenges. The exploration of efficient scientific medical educational methodologies has consistently been a focal point in the realm of residency training research. Numerous scholars have conducted extensive investigations into various aspects of medical education, including pedagogical strategies, content delivery, and instructional modalities, in an ongoing quest to identify more effective teaching methodologies [5–7].

Ultrasound medicine, recognized as a crucial medical imaging modality, holds increasing importance in clinical diagnosis, treatment, and monitoring, particularly within the domains of emergency medicine, oncology, and cardiovascular disorders. Furthermore, the utilization of ultrasound technology extends beyond traditional imaging assessments to encompass interventional therapies and real-time monitoring applications [8–10]. Unlike other radiographic techniques, ultrasound necessitates direct interaction with patients, often requiring positional adjustments and specific patient actions (such as

**Table 4. Satisfaction scores of the teaching mode questionnaire between the control group and experimental group (points, x±s).**

| item | control group | Observation group | t value | P value |
|---|---|---|---|---|
| Stimulate learning interest | 3.86±0.73 | 4.67±0.48 | 4.25 | 0.000 |
| Improve thinking and problem solving skills | 3.71±0.64 | 4.38±0.50 | 3.75 | 0.001 |
| Cultivate clinical thinking | 3.57±0.60 | 4.43±0.51 | 5.01 | 0.000 |
| Enhance communication skills | 3.38±0.50 | 4.76±0.44 | 9.56 | 0.000 |
| Enhance clinical learning confidence | 3.43±0.51 | 4.10±0.77 | 3.31 | 0.002 |
| Improve practical operation ability | 3.57±0.68 | 4.67±0.48 | 6.04 | 0.000 |
| Suitable for standardized training | 4.00±0.70 | 4.67±0.48 | 3.56 | 0.001 |

breath-holding, abdominal distension, and positional changes); the examination duration can be considerable, underscoring the significance of communication between ultrasound practitioners and patients.

In this study, we employed the Mini-CEX evaluation instrument to effectively assess the clinical competencies of resident physicians while providing timely feedback for improvement. The Mini-CEX assessment facilitates faculty members in gauging residents' foundational knowledge, procedural skills, clinical reasoning, and problem-solving capabilities. This method explicitly addresses the residents' weaker training areas, encourages self-reflection, and aids in overcoming training-related challenges, consequently enhancing their motivation and urgency for learning. In alignment with the competencies expected of residents [11], our findings indicate that the collaborative efforts of educators and residents resulted in marked improvements across all Mini-CEX metrics throughout the training period, significantly bolstering the residents' ultrasonic clinical proficiency. This innovative instructional approach proves to be effective.

Case-Based Learning (CBL) has emerged as a contemporary pedagogical method that has gained traction in recent years, optimizing the instructional process of ultrasonic diagnostics through the utilization of representative case sonograms. This approach effectively connects clinical cases with practical applications, transforming abstract theoretical concepts into tangible clinical practices, thereby enhancing students' analytical and problem-solving skills while deepening their comprehension of the material. In our case-based instruction, we employed Mind Maps to visually organize and synthesize knowledge pertinent to various diseases, encompassing anatomy, pathophysiology, clinical manifestations, laboratory assessments, ultrasonic diagnostics, and differential diagnoses. This technique effectively strengthens students' retention and mastery of case-related knowledge while fostering the systematic thinking abilities of resident physicians. The skills acquired through this instructional method are transferable to future academic and professional endeavors, further enhancing both learning and work efficacy. Li Jia [12], Andrea C Lorwald [13] et al. posited that the integration of the enhanced Mini-CEX, in conjunction with a DOPs dual-track evaluation framework, serves as a valuable assessment and teaching instrument in the clinical competency evaluation of ultrasound specialty residency training, yielding significant advancements in educational outcomes.The improvement of clinical skills within this domain presents significant advantages, thereby effectively enhancing the operational skill level of trainees and positively contributing to the overall quality of teaching. According to Sun Fang [14] et al, the Mini-CEX examination is characterized by its ease of administration and efficiency in time management. This assessment, grounded in professional theory and practical skill evaluations, emphasizes humanistic care, communication skills, and comprehensive clinical competence, with a particular focus on Assessment of professional skills. Palaniappan V [15] and Sajadi AS [3] indicate that the application of Mind Map teaching methods can significantly boost students' learning engagement, facilitate the formation of a clearer knowledge framework throughout the educational process, and augment their capacity for information retention and application. Rong XJ [16] et al. assert that students engaged in Case-Based Learning (CBL) demonstrate superior operational skills and theoretical knowledge compared to their peers instructed through conventional methodologies, alongside exhibiting elevated levels of learning satisfaction and self-assessment capabilities. Numerous studies have substantiated that demand-driven and workplace-oriented comprehensive assessment frameworks, including Mini-CEX, CBL, and Mind Map methodologies, can enhance students' overall satisfaction while simultaneously improving teaching effectiveness in a resource-efficient manner [6,7]. The traditional teaching mode is teacher-centered, that is, teachers mainly give lectures. Resident physicians are in a passive receiving position, lacking opportunities for active exploration and in-depth thinking, which is not conducive to the cultivation of independent thinking and problem-solving abilities. This study attempts to construct a new teaching model of ultrasound medicine, taking the model of "task-driven by the specific diseases required by the standardized training syllabus" as the orientation of clinical teaching of ultrasound medicine, and centering on the trainees. Based on specific diseases, guide resident physicians to think, research and summarize. Meanwhile, in the operation, pay attention to the cultivation of the humanistic care ability of resident physicians. Through evaluation, feedback and improvement, improve the overall job competence of resident physicians. The findings reveal that the integration of Mini-CEX with CBL and Mind Map methodologies significantly elevates performance in both theoretical and practical assessments when

compared to traditional instructional strategies. Furthermore, the results from the questionnaire survey indicate that this combined approach surpasses traditional teaching methods in terms of stimulating learning interest, enhancing critical thinking and problem-solving capabilities, nurturing clinical reasoning, improving communication skills, boosting confidence in clinical learning, refining practical operational proficiencies, and adapting to standardized training requirements.

However, this study was limited by a restricted sample size and was unable to distinguish the independent contributions of the three intervention measures. Future research needs to further decompose specific effects through factorial design. which precludes a comparative analysis of the learning outcomes between ultrasound students and their non-ultrasound counterparts. Additionally, the research duration was brief, hindering the tracking of students' clinical abilities in their subsequent professional endeavors. Moreover, the Mini-CEX assessment may be subject to subjective biases from instructors, with high expectations placed on teaching standards.

In conclusion, the integration of Mini-CEX, CBL, and Mind Map methodologies creates a novel training model for ultrasound residents. This multifaceted training approach not only enhances the clinical application skills of ultrasound medicine residents but also fosters their abilities for independent and lifelong learning, effectively strengthening their clinical competencies.

## Supporting information

**S1 File. Data.**
(ZIP)

## Acknowledgments

Thank you to the leaders, colleagues, and general training residents for their strong support in this research and paper writing.

## Author contributions

**Data curation:** Rong Liu, Ying Zhou, Lu Xiong.

**Investigation:** Ying Zhou, Lu Xiong.

**Project administration:** Rong Liu, Lei Li.

**Supervision:** Lei Li.

**Writing – original draft:** Rong Liu.

**Writing – review & editing:** Rong Liu.

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
