## [Decision Letter · Decision Letter 0]

Dear Dr. Liu,

Thank you for submitting your manuscript to PLOS ONE. After careful consideration, we feel that it has merit but does not fully meet PLOS ONE’s publication criteria as it currently stands. Therefore, we invite you to submit a revised version of the manuscript that addresses the points raised during the review process.

The ethical issues raised by Reviewers No 3 need special attention of the authors along with other comments of the reviewers.

We look forward to receiving your revised manuscript.

Kind regards,

Aamir Ijaz, MD, FCPS, FRCP, MCPS-HPE

Academic Editor

PLOS ONE

Journal Requirements:

Reviewers' comments:

Reviewer's Responses to Questions

**Comments to the Author**

1. Is the manuscript technically sound, and do the data support the conclusions?

Reviewer #1: Partly

Reviewer #2: Yes

Reviewer #3: No

2. Has the statistical analysis been performed appropriately and rigorously?

Reviewer #1: Yes

Reviewer #2: Yes

Reviewer #3: No

3. Have the authors made all data underlying the findings in their manuscript fully available?

Reviewer #1: Yes

Reviewer #2: Yes

Reviewer #3: Yes

4. Is the manuscript presented in an intelligible fashion and written in standard English?

Reviewer #1: Yes

Reviewer #2: Yes

Reviewer #3: Yes

Reviewer #1: 1. The title gives the impression that MiniCEX is part of the intervention employed. The intervention in this study is CBL and mind-maps. MiniCEX is used to evaluate outcomes in experimental and control groups. The authors should perhaps rephrase the title for clarity.

2.The methodology section requires some clarification.

a) The cohort in both groups has residents, social training participants and graduates (lines 70-74) The question of uniformity and standardization arises when looking at how these groups are structured and assessed as the have very obvious discordant baseline competence and knowledge base. Furthermore, the residents are a mix of senior and junior but are evaluated on the same rubric of MiniCEX.

b) The authors state that mind mapping and CBLs are incorporated (lines 90-96) for the experimental group while traditional teaching is done for the control group in the classroom. It is important to state that in clinical OPD settings, there was no difference or segregation in the schedules and methods in place, in terms of number and type of patients seen, for the purpose of uniformity.

c)The questionnaire used is supported by validity statistics in the results section (lines 114-116). There is no mention in methodology section of how it was piloted and validated.

d) The authors mention how soft skills are also inculcated in the residents (98-104). It should be stated that this domain is uniform for both groups.

e) There is no mention of type of study design

3.It is suggested that paraphrasing be done for the following terms used in the manuscript

''cultivation of clinical abilities'' (line 195-196)

''evaluating professional quality'' (line 202)

''observational group''

Reviewer #2: I have reviewed the article, and I have found it fit for publication. The various aspects that were required to be covered up for article writing have been completed. The domains of ethics, results, tables have been well covered.

Reviewer #3: This study appears to be an interventional study where one group of participants is receiving an intervention i.e. CBL and mind mapping and evaluation is being done by Mini-CEX whereas, the observation group is only receiving traditional training. Please note that:

1. It is unethical to intentionally deprive a group of learners from an established learning and assessment methods that ensure better learning as CBL, mind mapping, and Mini-CEX have an established role in the aforementioned areas. For this reason, in medical education context, it is preferred to have a single arm study or there should be a possibility of crossover of two groups.

2. There is no description of traditional method of training and evaluation that is being used for the control group.

In addition, Mini-CEX is traditionally used to measure seven clinical skills including: history taking, physical examination, communication skills, clinical judgement, professionalism, organization/efficiency and overall clinical judgement on a 9-point scale. Whereas, the Mini-CEX rubric used in this study seems to be a modified scale measuring medical history taking, ultrasound examination, doctor-patient communication, clinical reasoning, and organizational efficiency. If modifications have been done, the process of modifications and pilot testing is not there.

Lastly, with three interventions in one group in a single exposure have created uncertainty regarding existing results due to multiple confounding variables linked to CBL, mind mapping and Mini-CEX encounters. It is difficult to say that which of the intervention has led to improvement in scores.

**Do you want your identity to be public for this peer review?** For information about this choice, including consent withdrawal, please see our Privacy Policy

Reviewer #1: **Yes: ** Dr Zahra Rashid Khan

Reviewer #2: **Yes: ** RIZWAN HASHIM

Reviewer #3: **Yes: ** Syeda Amina Ahmad

---

## [Author Response · Author response to Decision Letter 1]

13 Jun 2025

Responses to reviewers' commentse

Dear reviewers:

Thank you very much for your comments and professional-advice. These opinions help to improve the academic rigor of our article. Based on your suggestion and request, we have made the corrected modifications to the revised manuscript. We hope that our work can be improved again. urthermore, we would like to show the details as follows:~

Responses to comments-from-Reviewer1:

Reviewer #1: 1. The title gives the impression that MiniCEX is part of the intervention employed. The intervention in this study is CBL and mind-maps. MiniCEX is used to evaluate outcomes in experimental and control groups. The authors should perhaps rephrase the title for clarity.

Author response: We appreciate your feedback regarding the title. This study was an intervention study. Participants in the experimental group received the intervention, namely CBL and Mind mapping was conducted, and assessment was completed using Mini-CEX, while the control group only received traditional training. We have detailed the experimental methods in the manuscript. Additionally, to emphasize the intervention effects of CBL and mind mapping, we have revised the titles accordingly.

2.The methodology section requires some clarification.

a) The cohort in both groups has residents, social training participants and graduates (lines 70-74) The question of uniformity and standardization arises when looking at how these groups are structured and assessed as the have very obvious discordant baseline competence and knowledge base. Furthermore, the residents are a mix of senior and junior but are evaluated on the same rubric of MiniCEX.

Author response: Thank you for your feedback. We have detailed the composition of the two groups of cohorts, the inclusion and exclusion criteria, and analyzed the differences in baseline ability and knowledge base in the revised paper to improve the validity and reliability of the evaluation. (lines 82-88)

b) The authors state that mind mapping and CBLs are incorporated (lines 90-96) for the experimental group while traditional teaching is done for the control group in the classroom. It is important to state that in clinical OPD settings, there was no difference or segregation in the schedules and methods in place, in terms of number and type of patients seen, for the purpose of uniformity.

Author response: We appreciate your insightful comment. In our revised manuscript, we have clarified that while the experimental and control groups experienced different teaching methodologies in the classroom, the clinical OPD settings maintained uniformity in terms of the schedules and types of patients seen. This information has been added to the methodology section to ensure clarity and transparency regarding the consistency of clinical exposure across both groups.(lines 94-100)

c)The questionnaire used is supported by validity statistics in the results section (lines 114-116). There is no mention in methodology section of how it was piloted and validated.

Author response: Thank you for pointing out this. We have supplemented the detailed information on the trial and validation process of the questionnaire in the methodology section, including the specific methods and results of validity statistics, to enhance the reliability of the study. (line 177

d) The authors mention how soft skills are also inculcated in the residents (98-104). It should be stated that this domain is uniform for both groups.

Author response:Thank you for pointing out this. In the revision, we further clarified the indoctrination method of soft skills and emphasized the consistency of this method in both groups to ensure that all participants could receive the same soft skills training. (lines 94-128)

e) There is no mention of type of study design

Author response: Thank you for your feedback. In the introduction part of the manuscript, we have added an explanation of the type of research design and clarified the design method adopted in this study, so that readers can better understand the framework and purpose of the research. (line 60

3.It is suggested that paraphrasing be done for the following terms used in the manuscript

Author response: We appreciate the reviewer's suggestion regarding the paraphrasing of specific terms. We have carefully reviewed the manuscript and made the necessary adjustments to ensure clarity and improve readability. The terms in question have been paraphrased to better convey the intended meaning while maintaining the integrity of the content.

''cultivation of clinical abilities'' (line 195-196)

Author response: Thank you for pointing this out. We have changed "cultivation of clinical competence" to "improvement of clinical skills". (line 230)

''evaluating professional quality'' (line 202)

Author response: Thank you for your feedback. We found that "evaluating Professional Qualities" had some issues; therefore, we changed it into "Assessment of Professional Skills." (lines 236-237)

''observational group''

Author response: We appreciate your feedback on this issue. In order to reflect the research design more accurately, all "observation groups" have been changed to "experimental groups".

Reviewer #2: I have reviewed the article, and I have found it fit for publication. The various aspects that were required to be covered up for article writing have been completed. The domains of ethics, results, tables have been well covered.

Author response: Thank you for your positive feedback regarding the suitability of our article for publication. We appreciate your acknowledgment of the various aspects we covered, including ethics, results, and tables.We are glad to hear that you found all necessary aspects of the article writing process to be addressed adequately. Your recognition of our efforts in this regard is greatly appreciated.Thank you for your commendation on our coverage of ethics, results, and tables. We made sure to adhere to the guidelines and standards in these areas to ensure the integrity and clarity of our research.

Reviewer #3: This study appears to be an interventional study where one group of participants is receiving an intervention i.e. CBL and mind mapping and evaluation is being done by Mini-CEX whereas, the observation group is only receiving traditional training. Please note that:

1. It is unethical to intentionally deprive a group of learners from an established learning and assessment methods that ensure better learning as CBL, mind mapping, and Mini-CEX have an established role in the aforementioned areas. For this reason, in medical education context, it is preferred to have a single arm study or there should be a possibility of crossover of two groups.

Author response: We appreciate your attention to the research design. In our study, we chose a single-group design to evaluate the combined effect of the three intervention measures, which helps provide preliminary data on the relative efficacy of the combined teaching methods directly compare their effectiveness with that of the traditional method, draw more definite conclusions on the effectiveness of these methods, and provide a basis for future research. We also acknowledge that future research should consider factorial design or two-group crossover design to further verify our findings. We appreciate your concern about the ethical implications of limiting learners' access to their existing educational methods. in our study, we ensured that all participants had access to basic learning resources and support (lines 94-100). Furthermore, the research design was approved by the Institutional Review Board. This approval considered the ethical factors of the study.

2. There is no description of traditional method of training and evaluation that is being used for the control group.

Author response:Thank you for your feedback. We have added a detailed description of the traditional method of training and evaluation used for the control group in the revised manuscript. This includes specific training protocols, evaluation metrics to provide clarity on the control group's methodology.�lines 100-107

In addition, Mini-CEX is traditionally used to measure seven clinical skills including: history taking, physical examination, communication skills, clinical judgement, professionalism, organization/efficiency and overall clinical judgement on a 9-point scale. Whereas, the Mini-CEX rubric used in this study seems to be a modified scale measuring medical history taking, ultrasound examination, doctor-patient communication, clinical reasoning, and organizational efficiency. If modifications have been done, the process of modifications and pilot testing is not there.

Author response: We appreciate your revision suggestions. Your observation is very important. We did modify the Mini-CEX scale to meet the specific needs of this study. We detailed the reasons and process of the revision in the revised draft.In our research, we made adjustments to the traditional Mini-CEX to better adapt to the specific context of our study. These modifications include focusing on medical history records, ultrasound examinations, doctor-patient communication, clinical reasoning and tissue efficiency.�lines 136-144

Lastly, with three interventions in one group in a single exposure have created uncertainty regarding existing results due to multiple confounding variables linked to CBL, mind mapping and Mini-CEX encounters. It is difficult to say that which of the intervention has led to improvement in scores.

Author response: Thank you for your feedback. We acknowledge the impact of the potential confounding variables related to the three intervention strategies. We analyzed the relative contributions of each intervention strategy in the discussion section. We believe that the combination of these strategies may lead to an increase in scores, instead of the result of a single intervention. In the revised manuscript, we included discussions that highlight the limitations of our research concerning this issue and emphasize the need for further studies to clarify the impact of each intervention strategy. This method enables us to provide a clearer explanation of the results, while acknowledging the complexity of the intervention strategies.(lines 248-259)(lines 268-270)

6.PLOS authors have the option to publish the peer review history of their article (what does this mean?). If published, this will include your full peer review and any attached files.

Do you want your identity to be public for this peer review? For information about this choice, including consent withdrawal, please see our Privacy Policy.

Reviewer #1: Yes: Dr Zahra Rashid Khan

Reviewer #2: Yes: RIZWAN HASHIM

Reviewer #3: Yes: Syeda Amina Ahmad

Author response: Yes:Rong Liu

---

## [Editor Report · Decision Letter 1]

The application of CBL and mind mapping combined with Mini-CEX teaching mode in the cultivation of clinical competence of ultrasound residents

PONE-D-25-16295R1

Dear Dr. Liu,

We’re pleased to inform you that your manuscript has been judged scientifically suitable for publication and will be formally accepted for publication once it meets all outstanding technical requirements.

Kind regards,

Aamir Ijaz, MD, FCPS, FRCP, MCPS-HPE

Academic Editor

PLOS ONE
---

## [Editor Report · Acceptance letter]

PONE-D-25-16295R1

PLOS ONE

Dear Dr. Liu,

I'm pleased to inform you that your manuscript has been deemed suitable for publication in PLOS ONE. Congratulations! Your manuscript is now being handed over to our production team.

Kind regards,

on behalf of

Professor Aamir Ijaz

Academic Editor

PLOS ONE